# Surgical Robot Transformer (SRT):
# Imitation Learning for Surgical Tasks

**Ji Woong Kim**[1]    **Tony Z. Zhao**[2]    **Samuel Schmidgall**[1]    **Anton Deguet**[1]
**Marin Kobilarov**[1]    **Chelsea Finn**[2]    **Axel Krieger**[1]

Johns Hopkins University[1]    Stanford University[2]

https://surgical-robot-transformer.github.io/

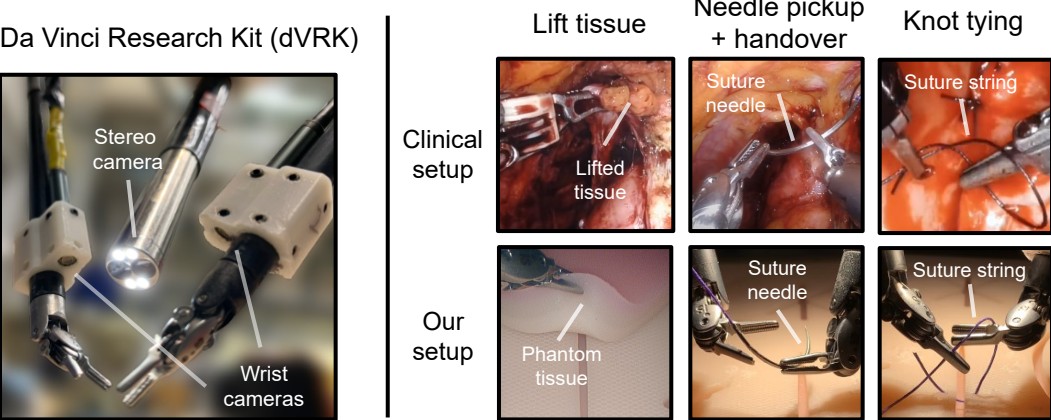

Figure 1: (*Left*): The da Vinci Surgical Research Kit (dVRK) system is equipped with a surgical endoscope and wrist cameras. (*Right*): Three fundamental surgical tasks are learned, including lift tissue (i.e. tissue retraction), needle-pickup and handover, and knot-tying which are among the most common surgical tasks.

**Abstract:** We explore whether surgical manipulation tasks can be learned on the da Vinci robot via imitation learning. However, the da Vinci system presents unique challenges which hinder straight-forward implementation of imitation learning. Notably, its forward kinematics is inconsistent due to imprecise joint measurements, and naively training a policy using such approximate kinematics data often leads to task failure. To overcome this limitation, we introduce a relative action formulation which enables successful policy training and deployment using its approximate kinematics data. A promising outcome of this approach is that the large repository of clinical data, which contains approximate kinematics, may be directly utilized for robot learning without further corrections. We demonstrate our findings through successful execution of three fundamental surgical tasks, including tissue manipulation, needle handling, and knot-tying.

**Keywords:** Imitation Learning, Manipulation, Medical Robotics

## 1  Introduction

Recently, large-scale imitation learning has shown great promise in creating generalist systems for manipulation tasks [1]. Prior research in this area has mostly focused on learning day-to-day household activities. However, an under-explored area with high potential is the surgical domain, particularly with the use of Intuitive Surgical's da Vinci robot. These robots are deployed globally and possess immense scaling potential: as of 2021, over 10 million surgeries have been performed using 6,500 da Vinci systems in 67 countries, with 55,000 surgeons trained on the system [2]. Often, the video and kinematics data are recorded for post-operative analysis, resulting in a large repository of

8th Conference on Robot Learning (CoRL 2024), Munich, Germany.

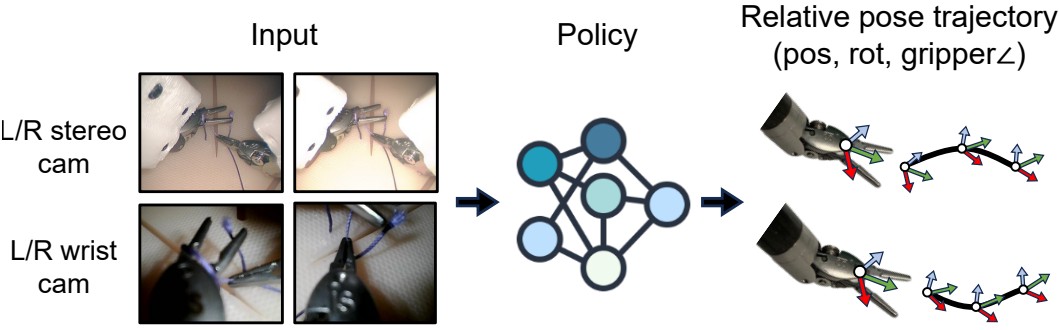

Figure 2: We propose a policy design which only takes images as input and outputs relative pose trajectories for both arms. Modeling policy actions as relative motion is a key ingredient that makes robot learning work on the dVRK.

demonstration data. Utilizing such large scale data holds significant potential for building generalist systems for autonomous surgery [3].

However, robot learning on the da Vinci presents unique challenges. The hardware suffers from inaccurate forward kinematics due to potentiometer-based joint measurements, hysteresis, and overall flexibility and slack in its mechanism [4]. These limitations result in the robot's failure to perform simple visual-servoing tasks [5]. As we discover in this work, naively training a policy using such approximate kinematics data almost always leads to task failure. For instance, a policy trained to output absolute end-effector poses, which is a common strategy to train robot policies, achieves near-zero success rates across all tasks explored in this work, including tissue lift, needle pickup and handover, and knot-tying (Fig. 1). To achieve robot learning at scale, we must devise a strategy that leverages such approximate kinematics data effectively.

Towards this end, we present an approach for robot learning on the da Vinci using its approximate kinematics data. Intuitively, our approach is based on the observation that the relative motion of the robot is much more consistent than its absolute forward kinematics. We thus model policy actions as differential motion and further explore its variants to design the most effective action representation for the da Vinci. We find that training an imitation learning algorithm using such relative formulation shows robustness to various configuration changes to the robot, even those known to significantly disrupt the robot's forward kinematics. Specifically, the da Vinci tools can be removed and reinstalled and all the robot joints can be freely moved, including the notoriously inaccurate set-up joints [4], without significantly impacting policy performance.

Additionally, we explore the use of wrist cameras in the surgical workflow. While not commonly employed in clinical settings, wrist cameras have demonstrated effectiveness in improving policy performance and facilitating generalization to out-of-distribution scenarios, such as varying workspace heights or unfamiliar visual distractions [6]. We thus evaluate their impact on performance and practical potential by designing removable brackets that enable easy sharing across various surgical instruments.

Overall, our results indicate that the relative motion on the da Vinci is more consistent than its absolute motion. Following this result, we further observe that a carefully chosen relative action representation can sufficiently train policies that achieve high success rates in surgical manipulation tasks. Additionally, using wrist cameras significantly improves policy performance, especially during phases of the procedure when precise depth estimation is crucial. In robustness tests, our model demonstrates the ability to generalize to novel scenarios, such as in the presence of unseen 3D suture pads and animal tissues, showing promise for future extensions into pre-clinical research.

Our main contributions are: (i) a successful demonstration of imitation learning on the da Vinci while using its approximate kinematics data and without requiring further kinematics corrections, while drastically outperforming the baseline approach; (ii) experiments showing that imitation learning can effectively learn complex surgical tasks and generalize to novel scenarios such as in the presence of

unseen realistic tissue; (iii) ablative experiments demonstrating the importance of wrist cameras for learning surgical manipulation tasks.

## 2 Related Work

**Manipulation and Imitation Learning**  Imitation learning enables robots to learn from expert demonstrations [7]. Behavioral cloning (BC) is a simple instantiation of imitation learning that directly predicts actions from observations. Early works tackle this problem through the lens of motor primitives [8, 9, 10, 11]. With the development of deep learning and generative modeling, different architectures and training objectives have been proposed to model the demonstrations end-to-end. This includes the use of ConvNets or ViT [12] for image processing [13, 14, 15], RNN or transformers for fusing history of observations [16, 17, 18], tokenization of the action space [19], generative modeling techniques such as energy-based models [20], diffusion [21] and VAEs [22, 23]. Prior works also focus on the few-shot aspect of imitation learning, [24, 25, 26, 27], language conditioning [15, 18, 19, 28], co-training [19, 29, 28], retrieval [30, 31, 32], using play data [33, 34, 35, 36], using human videos [37, 38, 39, 40, 41, 42], and exploiting task-specific structures [43, 44, 45].

However, most of these prior works focus on table-top manipulation in home settings. Surgical tasks, on the other hand, pose a unique set of challenges. They require precise manipulation of deformable objects, involve hard perception problems with inconsistent lighting and occlusions, and surgical robots may often have inaccurate proprioception and hysteresis [4] that are less pronounced in industrial arms. While in principle end-to-end imitation learning could capture these variations implicitly, it is unclear what design choices are important to enable effective learning in this regime.

We also note that in the dVRK community, the inaccuracies of the robot have been addressed via hand-eye calibration [46, 47, 48]. However, we assume that the demonstration dataset has already been collected without hand-eye calibration e.g., the large-scale clinical data, which means precise ground-truth kinematics is not available during training. Therefore, hand-eye calibration is not a fundamental solution to our problem.

**Autonomous Surgery**  Prior works in autonomous surgery primarily focus on designing task-specific policies for specific tasks, such as for suturing [49, 50, 51, 52], endoscope control [53, 54], navigation [55, 56, 57], and tissue manipulation [58, 59]. However, these methods do not typically scale well across various tasks or generalize well to varying environmental conditions. In contrast, end-to-end imitation learning offers a relatively simple solution to these shortcoming by only requiring good robot demonstrations. While prior works have also explored the use of imitation learning for surgical tasks [60, 61, 62, 55], its application to complex manipulation tasks like knot-tying remains unexplored, and practical design choices for implementing on the da Vinci have not been addressed.

## 3 Technical Approach

Consider the dVRK system, as illustrated in Fig. 3, which includes both the robot and a teleoperation console for user interaction. The dVRK features an endoscopic camera maipulator (ECM) and two patient side manipulators (PSM1, PSM2) sharing the same robot base. Each arm is a sequential combination of set-up joints (SUJ) which are passive, followed by active joints which are motorized (Fig. 3). The passive joints are notoriously inaccurate due to using only potentiometers for joint measurements. The active joints use both potentiometers and motor encoders, providing improved precision. However, in general, the use of potentiometers throughout all the joints causes the forward kinematics of the arms to be inaccurate, even up to 5cm error [4].

Using the dVRK console, the user collects many demonstrations of a task, acquiring a dataset $D = \{\tau_1, ..., \tau_N\}$, where each trajectory $\tau_i = \{(o_1, x_1, a_1), ..., (o_T, x_T, a_T)\}$ is a collection of observation $o_t$, proprioception $x_t$, and actions $a_t$, collected at time step $t$. Specifically, the observation $o_t$ includes left/right surgical endoscope images and left/right wrist camera images, totaling four images (Fig. 2), proprioception is the current pose of the PSMs w.r.t surgical endoscope tip frame denoted as $x_t = \{g_t^l, \ g_t^r\}$, where $g = (p, R) \in SE(3)$ and $\{l, r\}$ denotes the left and right grippers respectively, and actions are the commanded desired waypoints to reach specified via teleoperation control, denoted as $a_t = \{\hat{g}_t^l, \ \hat{g}_t^r\}$.

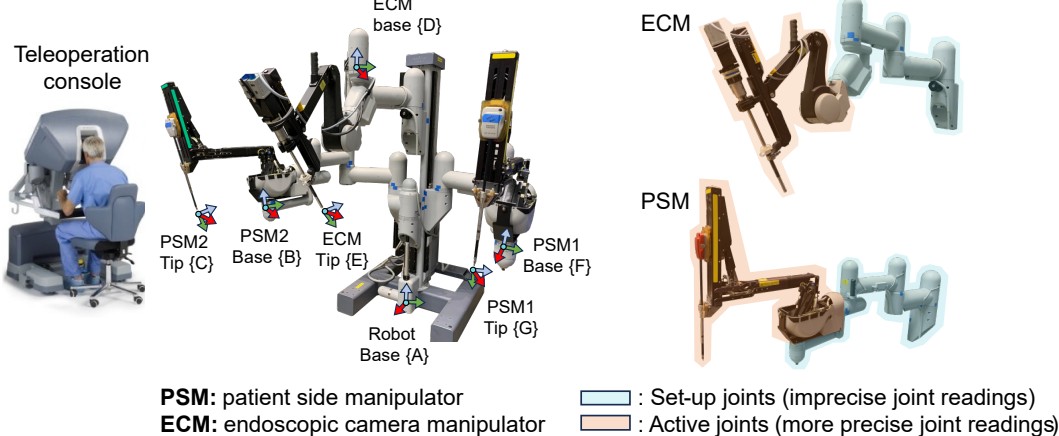

**PSM:** patient side manipulator
**ECM:** endoscopic camera manipulator

☐ : Set-up joints (imprecise joint readings)
☐ : Active joints (more precise joint readings)

Figure 3: The dVRK system consists of an endoscopic camera manipulator (ECM) and two patient side manipulators (PSM1, PSM2). Unfortunately, the dVRK arms are notorious for providing inconsistent forward kinematics. This is due to the setup joints (blue) only using potentiometers for joint measurements, which can be unrelible. The active joints (pink) use both potentiometers and motor encoders, improving precision.

Our objective is to learn surgical manipulation tasks via imitation learning. Given the robot's inaccurate forward kinematics, choosing the appropriate action representation is crucial. To illustrate this, we investigate three action representations: camera-centric, tool-centric, and hybrid-relative as shown in Fig. 4. The camera-centric approach serves as a baseline, highlighting the limitations of modeling actions as absolute poses of the end-effectors. The tool-centric approach offer an improved formulation by modeling actions as relative motion, leading to higher success rates. The hybrid-relative approach further improves beyond tool-centric approach by modeling translation actions with respect to a fixed reference frame, further improving accuracy in translation movements. These approaches are detailed below:

1. ***Camera-centric actions***: We model camera-centric actions as absolute poses of the end-effectors w.r.t the endoscope tip frame. The setup is similar to how position-based visual servoing applications (PBVS) are implemented and is a natural choice on the dVRK. Specifically, the objective is to learn a policy $\pi$ that, given an observation $o_t$ at time $t$, predicts an action sequence $A_{t,C} = (a_t, ..., a_{t+C})$, where $C$ denotes the action prediction horizon. The policy can thus be defined as $\pi : o_t \mapsto A_{t,C}$. This formulation is visually shown in Fig. 4.

2. ***Tool-centric actions***: We model tool-centric actions as relative motion w.r.t the *current* end-effector frame, which is a moving body frame. Tool-centric actions can be defined as:

$$A_{t,C}^{tool} = \left\{ (g_t^i)^T \hat{g}_s^i \mid s \in [t, t+C]; \ i \in \{l, r\} \right\} \tag{1}$$

Intuitively, the desired poses $\hat{g}_s^i$ are subtracted by the current end-effector poses $g_t^i$ using the $SE(3)$ subtraction rule, for each time up to horizon $C$ and for each corresponding left and right grippers. There are largely two benefits to adopting this action representation. A relative motion formulation is used, which we show later in Section 5, is more consistent compared to the absolute forward kinematics of the arms. Also, the subtraction cancels out the endoscope forward kinematics terms and the actions can be expressed in terms of the PSMs forward kinematics only. This effectively reduces the margin for errors since less joints are involved in representing actions. However, one caveat of this approach is that the delta motion is defined w.r.t a moving reference frame. This requires the policy to implicitly localize the current end-effector orientation from image observations, and output translations and rotations along the localized principal axes, which can be a challenging task. The policy objective can be defined as $\pi : o_t \mapsto A_{t,C}^{tool}$. This formulation is visually shown in Fig. 4.

3. ***Hybrid Relative Actions***: Similar to tool-centric actions, hybrid relative actions are modeled as relative motion but w.r.t two different reference frames. Specifically, the delta translations are

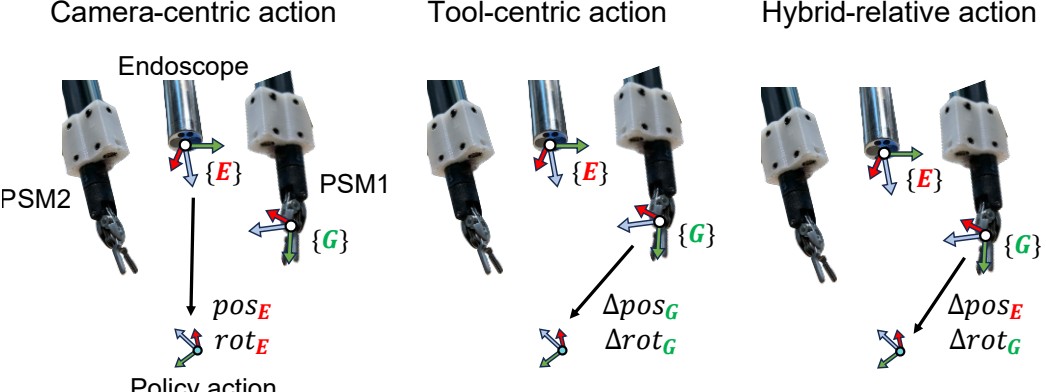

Figure 4: We consider three options for modeling policy actions. *(Left):* Camera-centric approach models actions as absolute end-effector poses w.r.t the endoscope tip frame. *(Middle):* Tool-centric approach models actions as delta positions and delta rotations defined w.r.t the current end-effector frame. *(Right):* Hybrid relative approach models actions as delta positions defined w.r.t the endoscope tip frame and delta rotations defined w.r.t the current end-effector frame.

defined w.r.t endoscope tip frame and delta rotations are defined w.r.t the current end-effector frame. This formulation can be defined as follows:

$$A_{t,C}^{hybrid} = \left\{ \hat{g}_s^i \ominus g_t^i \mid s \in [t, t+C];\ i \in \{l, r\} \right\} \tag{2}$$

Where the subtraction operation $\ominus$ defined as:

$$\hat{g}_s^i \ominus g_t^i = \left( \hat{p}_s^i - p_t^i, (R_t^i)^T \hat{R}_s^i \right) \tag{3}$$

Intuitively, the subtraction is performed between the corresponding translation and rotation elements i.e. vector subtraction for positions and $SO(3)$ subtraction for rotations. A key distinction of this approach from the tool-centric approach lies in modeling the delta translation with respect to the fixed frame of the endoscope-tip, rather than the moving frame of the end-effector. This approach removes the burden for the policy to localize the end-effector's orientation to generate delta translations along the localized axes, thereby improving the quality of translation motion. The policy can be defined as $\pi : o_t \mapsto A_{t,C}^{hybrid}$. This formulation is visually shown in Fig. 4.

## 4 Implementation Details

To train our policies, we use action chunking with transformers (ACT) [23] and diffusion policy [63]. The policies were trained using the endoscope and wrist cameras images as input, which are all downsized to image size of $224 \times 224 \times 3$. The original input size of the surgical endoscope images were $1024 \times 1280 \times 3$ and the wrist images were $480 \times 640 \times 3$. Kinematics data is not provided as input as commonly done in other imitation learning approaches because it is generally inconsistent due to the design limitations of the dVRK. The policy outputs include the end-effector (delta) position, (delta) orientation, and jaw angle for both arms. We leave further specific implementation details in Appendix A.

## 5 Experiments

In our experiments, we aim to understand the following key questions: (1) is imitation learning sufficient to learn challenging surgical manipulation tasks? (2) Is the dVRK's relative motion more consistent than its absolute forward kinematics? (3) Are using wrist cameras critical to achieving high success rates? (4) How well does our proposed model generalize to unseen novel scenarios? To answer these questions, we compare the task success rates of the policies trained using various action representations. We also directly compare the consistency of relative versus absolute motion by tracking a reference trajectory using the various action representations and comparing their tracking

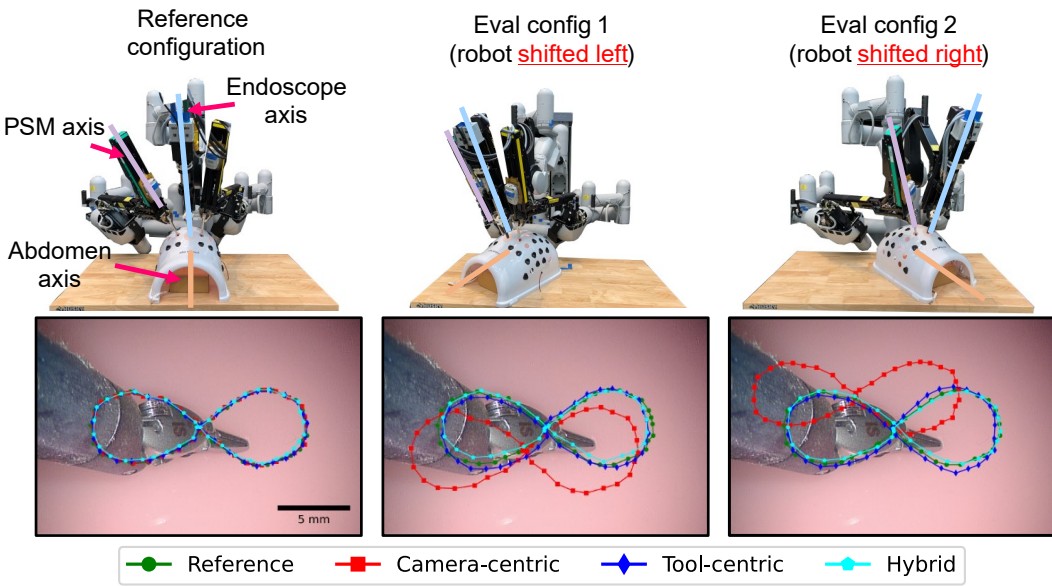

Figure 5: The repeatability of all action representations are tested by repeating a recorded reference trajectory under various robot configurations. (*Left*): The first column shows perfect reconstruction of the reference trajectory for all action representations since the robot joints have not moved since when the reference trajectory was collected. *(Middle, Right)* When the robot is shifted to the left or to the right, the camera-centric action representation fails to track the reference trajectory while the relative action representations track them quite closely. This is primarily due to the set-up joints being moved, which causes significant joint measurement errors. This experiment proves that in the presence of inconsistent joint measurements, relative motion can be more consistent.

Table 1: Trajectory tracking RMSE (mm) under various robot configurations

|  | Ref config | Eval config 1 | Eval config 2 |
|---|---|---|---|
| Camera-centric | 0.6 | 1.9 | 2.8 |
| Tool-centric | 0.9 | 0.9 | 0.7 |
| Hybrid-relative | 0.9 | 0.8 | 0.8 |

errors. We also explore the importance of wrist cameras by comparing the policy performance with and without them. Finally, we consider whether the proposed models can generalize to novel unseen scenarios, such as in the presence of animal tissues. These experiments are explored in the context of three tasks: lift tissue, needle pickup and handover, and knot-tying.

**Experiment Setup**    During data collection, the robot is set up in a reference configuration as shown in Fig. 5. In this configuration, 224 trials were collected for tissue lift, 250 trials for needle pickup and handover, and 500 trials for knot-tying, all collected by a single user across multiple days. During all experiments, a dome simulating the human abdomen (Fig. 5) was used to roughly place the arms and the endoscope in an approximately similar location using the same holes. The placement is only approximate because the holes are much larger than the endoscope and tool shaft size, and the tools have to be manually placed into the holes by moving the set-up joints.

**Evaluating the Consistency of Relative Motion vs. Absolute Forward Kinematics**    In this section we seek to understand whether relative motion on the dVRK is more consistent than its absolute forward kinematics. To test our hypothesis, we teleoperate a reference trajectory e.g., an infinity sign as shown in Fig. 5. This trajectory is then represented in various action representations using the formulas presented in Section 3. Then, we place the end-effector in the same initial pose and replay the trajectories using the various action representations under different robot configurations. These different configurations include shifting the robot workspace to the left and to the right side (Fig. 5). These workspace shifts cause the robot set-up joints to move, which are the joints prone to cause large joint measurement errors due to using only potentiometers for joint measurements. To

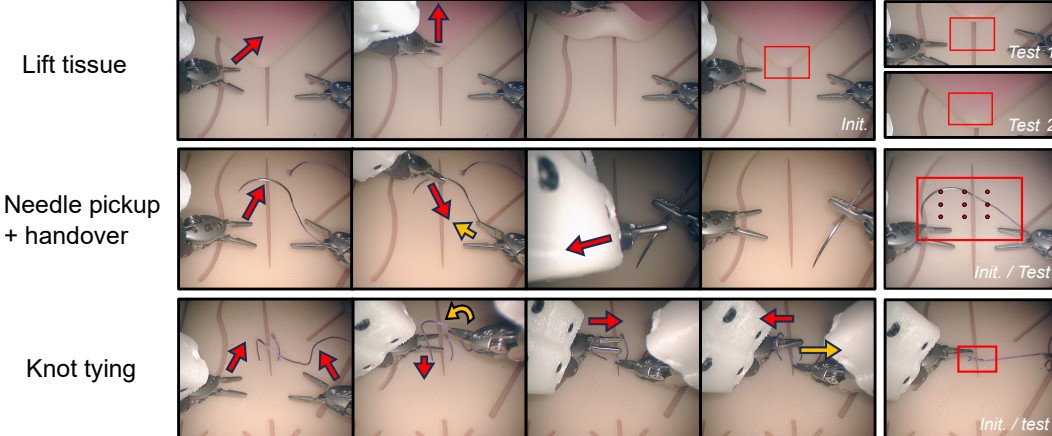

Figure 6: *(Top):* Tissue lift task requires grabbing the corner of the rubber pad (i.e. tissue) and lifting it upwards. During training the corner is kept within the red box and the configuration of the corners at test time is shown. *(Middle):* Needle pickup and handover is self-explanatory. The needle was placed randomly inside the red box during training. At test time, the center hump of the needle was placed at nine locations as shown, to enforce consistent setup during evaluation. *(Bottom):* Knot-tying requires creating a loop using the left string, grabbing the terminal end of the string through the loop, and pulling the grippers away from each other. During training, the location of the strings originating from the pads were randomly placed inside the red box, and at test time, it was centered in the red box as shown.

Table 2: Success rates on three surgical tasks using various action representations

| | | Tissue lift | | Needle pick + handover | | Knot tying | | |
| --- | --- | --- | --- | --- | --- | --- | --- | --- |
| | | Test 1 | Test 2 | Grasp | Handover | Grasp String | Loop | Whole Task |
| ACT [23] | Camera-centric | 0/5 | 0/5 | 0/9 | 0/9 | 0/20 | 0/20 | 0/20 |
| | Tool-centric | **5/5** | **5/5** | **9/9** | 5/9 | **20/20** | **20/20** | **18/20** |
| | Hybrid-relative | **5/5** | **5/5** | **9/9** | **9/9** | **20/20** | **20/20** | **18/20** |
| | Hybrid-relative (no wrist cam) | - | - | **9/9** | 6/9 | 8/20 | 4/20 | 4/20 |
| | Hybrid-relative (pork backgrd) | - | - | **9/9** | **9/9** | - | - | - |
| Diffusion Policy [21] | Hybrid-relative | **5/5** | **5/5** | 8/9 | 4/9 | 10/20 | 7/20 | 4/20 |

compare their tracking errors, the replayed trajectories are annotated at the end-effector (i.e. control point) in image coordinates and plotted, as shown in the bottom row of Fig. 5.

The plots in Fig. 5 and the numeric RMSE results in Table 1 show that in the reference configuration, all action representation precisely reconstruct the reference trajectory, since the set-up joints have not yet moved. However, when the robot configuration is changed by moving the set-up joints and new erroneous joint measurments are obtained, the camera-centric action representation fails to reconstruct the reference trajectory. Also, this error is not consistent for different robot configurations as shown in the trajectory plots in Fig. 5. For relative action representations, which include tool-centric and hybrid-relative action formulations, the reference trajectory is repeated more consistently, and their numeric errors do not vary significantly as observed in Table 1. In summary, this experiment shows that relative motion on the dVRK is more consistent compared to its absolute forward kinematics in the presence of inconsistent joint measurement errors.

**Policy Performance Using Various Action Representations**  We evaluate the policy performance using the various action representations on tissue lift, needle pickup and handover, and knot-tying as shown in Table 2. The camera-centric action representation performed poorly across all three tasks.

Because the joint measurements of the dVRK are inconsistent, the end-effectors almost always failed to reach the target objects (e.g., tissue corner, needle, and string) and often dangerously collided with the underlying rubber pads. The policy trained using tool-centric action representation showed improved performance across all three tasks. However, during needle pickup + handover when large rotations were involved, the handover phase of the task often failed. In particular, after picking up the needle, the left gripper had to make a $\sim 90$ degree rotation to transfer the needle to the opposing arm (Fig. 6). During this phase of the motion, the orientations of the grippers appeared correct, however, the translation motion appeared incorrect and seemed to be the cause of task failure. We conjecture this reason was due to grounding the policy actions to a moving end-effector frame. The policy is required to localize the moving end-effector orientation using image observations and generate delta translations along the localized principal axes of direction, which can be a challenging task. To fix this issue, the hybrid relative motion action was used, which grounds the translation motion to a fixed frame of the endoscope-tip. This formulation improved the translation errors during the aforementioned needle handover phase and ultimately achieved the highest success rates across all three tasks. This best performing action representation was also implemented on Diffusion Policy, however, the performance was not as high as ACT. We also observed that Diffusion Policy often generated unsmooth motions, potentially due to not being able to adapt to the inconsistent kinematics of the robot.

**Evaluating the Importance of Wrist Camera**   We also evaluate the importance of adding wrist cameras and their contribution to task success. To demonstrate this, we trained policies without wrist cameras on the needle pickup and handover and knot-tying tasks using the hybrid relative action formulation (Table 2). Overall, we observed that omitting wrist cameras lead to significant drop in performance. We conjecture that wrist cameras aid in scenarios where precise depth estimation is necessary. For instance, during the needle pickup and handover task, specifically during the latter phase transferring the needle, the wrist views clearly showed whether the needle was being navigated into the opposing grippers from afar. This additional view may have provided better context for task success. However, we observed that this level of information was difficult to discern from the third-person endoscopic view alone.

**Evaluating Generalization**   We also evaluate the ability of our models to generalize to novel scenarios, such as under more clinically relevant background using animal tissues (pork and chicken) and an unseen 3D suture pad. Most of this evaluation remains qualitative and greater details are elaborated in Appendix B. In terms of quantitative results, we evaluate the hybrid-relative action formulation on the needle pick-up and handover task on a pig loin background. We observe that its overall success rate is quite high (Table 2), however, the quality of the motion and the accuracy of the needle grasps were much lower compared to those observed in the core experiments. In terms of qualitative results, we observe multiple instances of successful knot-tying achieved on pork tissue, and needle grasps on chicken background and on an unseen 3D pad (Appendix B).

## 6   Limitations and Conclusion

In this work, we opt for using off-the-shelf large wrist cameras which are not clinically relevant. However, the cameras may be replaced with much smaller ones (1-2mm diameter) and its mount can be further optimized by integrating quick-release mechanisms for swift transfer between surgical tools. Also, our model is limited as it can only act based on current observations and does not have the ability to modulate different behavior based on human instruction. We hope to address these issues in future work to further advance the autonomy of surgical robots.

In summary, we demonstrated an approach for imitation learning on the dVRK using its approximate kinematics data, without providing further post-processing corrections. The key idea of our approach was to rely on the more consistent relative motion of the robot, achieved by modeling policy actions as relative motion such as tool-centric and hybrid-relative actions. As mentioned in the introduction, we believe that our work is a step towards leveraging the large repository of approximate surgical data for robot learning at scale, without providing further kinematics corrections. We believe more research in this direction can further guide the path towards building general-purpose systems towards autonomous surgery.

# 7 Acknowledgement

This material is based upon work supported by NSF/FRR 2144348, NIH R56EB033807, NSF DGE 2139757, and ARPA-H AY1AX000023.

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

# A  Implementation Details

For ACT, main modifications include changing the input layers to accept four images, which include left/right surgical endoscope views and left/right wrist camera views. The output dimensions are also revised to generate end-effector poses, which amounts to a 10-dim vector for each arm (position [3] + orientation [6] + jaw angle [1] = 10), thus amounting to a 20-dim vector total for both arms. The orientation was modeled using a 6D rotation representation following [21], where the 6 elements corrrespond to the first two columns of the rotation matrix. Since the network predictions may not generate orthonormal vectors, Gram-Schmidt process is performed to convert them to orthonormal vectors, and a cross product of the two vectors are performed to generate the remaining third column of the rotation matrix. For diffusion policy, similar modifications are made such as changing the input and the output dimensions of the network appropriately. The specific hyperparameters for training are shown in Table 3 and 4. Additionally, ACT was trained for approximately 12 - 15 hours and Diffusion Policy was trained for  24 hours for all tasks on an RTX 3090.

| | |
|---|---|
| learning rate | 1e-5 |
| batch size | 8 |
| # encoder layers | 4 |
| # decoder layers | 7 |
| feedforward dimension | 3200 |
| hidden dimension | 512 |
| # heads | 8 |
| chunk size | 100 |
| beta | 10 |
| dropout | 0.1 |

Table 3: Hyperparameters of ACT.

| | |
|---|---|
| learning rate | 1e-4 |
| batch size | 64 |
| chunk size | 32 |
| scheduler | DDIM[64] |
| train and test diffusion steps | 100, 100 |
| ema power | 0.75 |
| backbone | ResNet18[65] |
| noise predictor | UNet[66] |
| image augmentation | RandomCrop(ratio=0.95) & ColorJitter(brightness=0.3, contrast=0.4, saturation=0.5) & RandomRotation(degrees=[-5.0, 5.0]) |

Table 4: Hyperparameters of Diffusion Policy.

# B  Generalization to Novel Settings

Needle pickup and handover

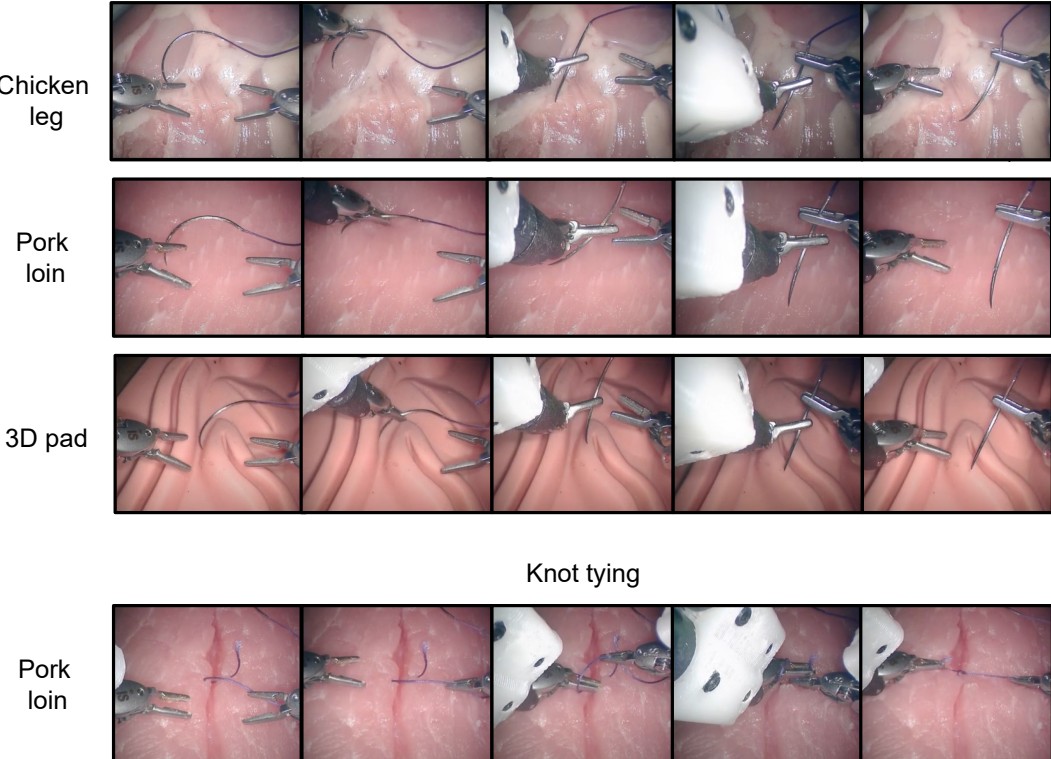

Knot tying

Figure 7: We show qualitative examples of our model generalizing to novel scenarios beyond training settings. *(Top):* Successful zero-shot needle pickup and handover on chicken leg, pork loin, and on a 3D pad. *(Bottom):* Successful zero-shot knot-tyng on pork loin.

