# OpenReview forum: "Surgical Robot Transformer (SRT): Imitation Learning for Surgical Tasks"
_robot-learning.org/CoRL/2024/Conference — CoRL 2024_

### Official Review · Reviewer_kUC3 · 2024-07-18
**Great work! Small step in IL giant leap on robot.**

**Originality:** 5
**Technical Quality:** 4
**Clarity Of Presentation:** 5
**Potential Impact:** 4
**Recommendation:** 4
**Confidence:** 4

**Review:**

# Originality

Though there is not too much novelty in the learning method as well as the data processing pipeline, the paper makes great contribution to the community studying several important setup in the camera and contributing an imitation learning dataset.

As the "large robotics models" are promising and prevailing in the manipulation tasks, we are looking forward for the migration of the idea to, I would say, more important task. And this paper makes that step out.

# Clarity

The paper is well-written.

# Strengths

1. The paper experiments different settings (action representation and model architecture) and conduct real world experiments.
2. Appreciate the "single user" who collected the dataset (Line 183). A great contribution to the community.
3. The paper conducts real-world evaluation of the trained model.


# Weaknesses

1. I would expect more discussion on the potential risks and solutions of current method. Connecting to Question 1, do we have any way to estimate the risk and safe capability of a trained model?
2. The technical part of the paper might not be novel. That is, the learning algorithm as well as the model architecture are not the focus of this paper.

**Quality Of The Limitations Section:**

2

**Questions For Rebuttal:**

1. Do we have fine-grained metrics in addition to the success rate? For example the "progress" the robot achieves so far before failure.
2. Have you studied the scalability of the model? That is, have you try to increase / decrease the scale of the model and see if it works? I think that would be very useful to the community to see, given current amount of data, what's the largest possible size of the model.

**Robotics Focus:**

4

**Summary Of Paper:**

The paper studies the imitation learning setting (i.e. the best action representation) of Da Vinci Research Kit on surgical tasks.

**Summary Of Recommendation:**

The paper is well structured and studied several settings in the surgical tasks. An important and bold attempt to apply large model toward surgical domain.

---

### Official Review · Reviewer_q2qZ · 2024-07-20
**Surgical Robot Transformer (SRT): Imitation Learning for Surgical Subtasks**

**Originality:** 2
**Technical Quality:** 3
**Clarity Of Presentation:** 4
**Potential Impact:** 3
**Recommendation:** 3
**Confidence:** 2

**Review:**

Strengths
* This paper is clearly written and proposes a straightforward method for an important class of manipulation problems.
* The hypothesis is clearly stated, notably that action representation is important given the noisy kinematic data of the robot
* The authors perform hardware experiments on the da Vinci platform. Aside from the speed of the videos, the demonstrations of dexterous manipulation are impressive.
* The authors' approach seems to generalize to unseen tasks and tissue types.

Weaknesses
* The only novel technical contribution appears to be changing the action representation to account for relative motion of the robot
* Although the method appears to work, the videos don’t demonstrate much diversity in the task. It would be much more informative if it were possible to discern the differences in the initial conditions of a particular task.
* The videos are sped up 4-12x, suggesting that the robot is moving quite slowly.
* It's unclear how much the performance of the robot could be improved by performing system identification and learning a better predictive model of the robot.

**Quality Of The Limitations Section:**

3

**Questions For Rebuttal:**

* Could the authors clarify if there is any novelty in their method beyond comparing the action representations?
* It would be interesting to know how much data is required to get good policies. The authors report ~250 expert demonstrations. Could they get away with less?
* The authors use ACT and Diffusion Policy but don’t mention how long either policy takes to train. They also don’t discuss inference times. This would be helpful in evaluating the method.

**Robotics Focus:**

4

**Summary Of Paper:**

The authors propose an imitation learning method to solve tasks that are relevant to robotic surgery such as suturing, tissue lift, and needle pickup. One of the major challenges with imitation learning on the da Vinci robot is dealing with the noisy forward kinematics. The authors hypothesize that modeling the action representation as relative motion of the robot might have better success across all tasks. To this end, the author compare camera-centric, tool-centric, and hybrid-relative actions to see which performs best.

**Summary Of Recommendation:**

I suggest a weak accept because while I interpret a lack of novelty in the proposed approach, the authors have succesfully demonstrated their approach working on real hardware. I believe this can be a very important step in doing imitation learning for surgical robotics.

---

### Official Review · Reviewer_XUen · 2024-07-21

**Originality:** 4
**Technical Quality:** 5
**Clarity Of Presentation:** 5
**Potential Impact:** 3
**Recommendation:** 4
**Confidence:** 3

**Review:**

**Strengths**
- The proposed approach is simple, intuitive, and well-described
- The proposed approach could enable further research in learning complex surgical tasks from existing datasets despite containing approximate kinematics.
- Experiments are convincing and demonstrate that the proposed method is effective and provides justification for the design choices based on the inferior performance of ablative baselines.
- Clinical demonstrations are very impressive to see. It would have been further improved by reporting even minimal statistics (e.g., success rate out of five attempts).

**Weaknesses and questions**
- The performance improvement over tool-centric design seems to be minimal, except in one stage of one task.
- The diffusion policy was not trained using the baseline representations, leaving out the possibility that the claimed benefits might diminish when a diffusion policy is used. Any justifications for why this is unlikely would be helpful.
- Why was the ablation without wrist camera only evaluated on the needle handover task?

**Quality Of The Limitations Section:**

2

**Questions For Rebuttal:**

Please address the questions and concerns from above.

**Robotics Focus:**

4

**Summary Of Paper:**

The paper proposes relative pose as an effective action representation to learn complex surgical tasks can be from demonstrations that have imperfect/noisy kinematics data.

**Summary Of Recommendation:**

The paper presents a simple yet effective action representation to enable learning complex surgical skills from existing datasets containing approximate kinematics. The paper is well-written, and the experimental results sufficiently demonstrate its effectiveness and justify the core design choices.

---

### Author Rebuttal · Authors · 2024-08-14

Dear reviewers,

We thank you for the constructive feedback of our work. Please see attached PDF for the revised draft. The revised sections have been highlighted in red.

To address the reviewers' questions, we have included the following new content in the revised manuscript:

- Additional no-wrist camera experiment for knot-tying in Table 2
- Training / inference times added to Appendix A
- Clarifications regarding system ID / more references added to the related works section
- Minor wording changes / figure revisions for better clarity across the document.

---

### Decision · Program_Chairs · 2024-09-04

**Decision:**

Accept

**Comment:**

The reviewers all agree that the manipulation capabilities of the presented work in real-world surgical experiments are impressive. However, there are concerns regarding the technical novelty of this approach as it seems to be limited to the action representation.
A clarification on the novelty of the presented approach and the action representation would benefit the paper.

Additional discussion on the potential risk of the approach would further strengthen the paper.
One potential risk is the smoothness of the action sequences produced by diffusion policies (even when combined with action chunking).
This issue has been extensively studied in previous work[1]. The paper would be further strengthened by discussing and comparing to this approach.

[1]: Scheikl et al, “Movement Primitive Diffusion: Learning Gentle Robotic Manipulation of Deformable Objects”, 2023

## Post Rebuttal
The rebuttal addressed all remaining concerns and further improved the already strong submission. The reviewers unanimously agree that this work should be published at CoRL 2024.